# Grading Evolution and Contemporary Prognostic Biomarkers of Clinically Significant Prostate Cancer

**DOI:** 10.3390/cancers13040628

**Published:** 2021-02-05

**Authors:** Konrad Sopyllo, Andrew M. Erickson, Tuomas Mirtti

**Affiliations:** 1Research Program in Systems Oncology, Faculty of Medicine, University of Helsinki, 00014 Helsinki, Finland; konrad.sopyllo@helsinki.fi; 2Nuffield Department of Surgical Sciences, University of Oxford, Oxford OX3 9DU, UK; andrew.erickson@nds.ox.ac.uk; 3Department of Pathology, HUS Diagnostic Centre, Helsinki University Hospital, 00029 Helsinki, Finland

**Keywords:** prostate cancer, survival, biomarker, Gleason grading, grade grouping

## Abstract

**Simple Summary:**

Prostate cancer treatment decisions are based on clinical stage and histological diagnosis, including Gleason grading assessed by a pathologist, in biopsies. Prior to staging and grading, serum or blood prostate-specific antigen (PSA) levels are measured and often trigger diagnostic examinations. However, PSA is best suited as a marker of cancer relapse after initial treatment. In this review, we first narratively describe the evolution of histological grading, the current status of Gleason pattern-based diagnostics and glance into future methodology of risk assessment by histological examination. In the second part, we systematically review the biomarkers that have been shown, independent from clinical characteristics, to correlate with clinically relevant end-points, i.e., occurrence of metastases, disease-specific mortality and overall survival after initial treatment of localized prostate cancer.

**Abstract:**

Gleason grading remains the strongest prognostic parameter in localized prostate adenocarcinoma. We have here outlined the evolution and contemporary practices in pathological evaluation of prostate tissue samples for Gleason score and Grade group. The state of more observer-independent grading methods with the aid of artificial intelligence is also reviewed. Additionally, we conducted a systematic review of biomarkers that hold promise in adding independent prognostic or predictive value on top of clinical parameters, Grade group and PSA. We especially focused on hard end points during the follow-up, i.e., occurrence of metastasis, disease-specific mortality and overall mortality. In peripheral blood, biopsy-detected prostate cancer or in surgical specimens, we can conclude that there are more than sixty biomarkers that have been shown to have independent prognostic significance when adjusted to conventional risk assessment or grouping. Our search brought up some known putative markers and panels, as expected. Also, the synthesis in the systematic review indicated markers that ought to be further studied as part of prospective trials and in well characterized patient cohorts in order to increase the resolution of the current clinico-pathological prognostic factors.

## 1. Introduction

Prostate cancer (PCa) diagnosis is based on needle biopsy histological assessment. The current globally accepted grading system, Gleason grading has been a product of prospective studies searching for prognostic features. The Gleason score (GS) is based on two complementary grade patterns that relate strictly to glandular architecture of the neoplastic tissue. As its foundation has held through the decades, some relevant changes to the grading criteria have been introduced during the last 15 years. The accumulating research on the significance of different grade patterns and morphologies has been consolidated into consensus criteria of reporting the grades and into the novel Grade grouping (GG) system, which simplifies the message and correlates even better with prognosis in many aspects. As we review the evolution of Gleason grading and the most important contemporary criteria of reporting of prostate tissue samples, we also review the current advances in quantitative and image analysis-driven cancer grading.

Although Gleason grading is considered the most significant prognostic factor in localized PCa, it does not inherently hold information on the biological features or genomic changes of the cancer. As a result of Gleason grading evolution, most of the new PCa diagnoses fall into GS 7 or GG 2/3, thereafter leaving the treating clinician with a need for further risk stratification and prognostic assessment. As one third of the patients experience so-called biochemical recurrence, i.e., BCR (PSA-levels increasing after initial treatment) during follow-up, BCR is considered only as a surrogate endpoint to metastasis or mortality later along the course of the disease. We have systematically in this report reviewed the published literature on putative biomarkers for clinically relevant endpoints, namely occurrence of metastases, PCa-specific death or death of any cause, in localized PCa. Emphasis is put into independent prognostic significance of the markers, most of which were found to be tissue biomarkers, in order to delineate their potential in aiding in risk assessment in addition to the conventional risk parameters. Some of the markers we found are reviewed in more detail in this Special Issue [1]. The publications on biomarkers with the strongest independent association with the abovementioned hard end-points are further explained in context of the source of the biomarker, whether being pre-treatment biopsy, surgical specimen, or peripheral blood.

## 2. Historical Development of Prostate Cancer Grading

Histopathological diagnosis of PCa stems from the first bright light microscopy examination and description of abnormal changes by J. Adams, a surgeon who worked at the London Hospital, in 1853 [2]. The first report concerning correlation of the histological state and the malignant potential of the disease was published by Broders in 1925. He invented a four-tier grading system that was based on the ratio of undifferentiated cells to the differentiated ones [3,4].

The basis for the current PCa grading system was introduced in 1966 by Donald F. Gleason. The system was created on the basis of a study performed in the years 1959–1964 by the Veteran’s Affairs Cooperative Research Group (VACURG), a research group set up by the lead urologist George T. Mellinger at the Minneapolis Veteran’s Hospital. Two hundred and seventy men were enrolled and the material used in the study was acquired from transurethral resections, biopsies, and radical prostatectomies (RP). Instead of cytology, a range of glandular architectural grade patterns was applied to assign a combined Gleason score, which was a product of a statistical approach showing that two grades combined are more accurate in outcome prediction than using only one grade pattern. The group’s study demonstrated a correlation between the increase in cancer specific-mortality and an increase in the histological score [5]. Gleason’s grading system was later validated by Gleason and Mellinger to have predictive value for disease specific mortality in a study made on 1032 patients [6]. Subsequently, several grading systems were compared in workshops organized by the American Cancer Society, and the Gleason grading system was recommended for broader use. Until this day Gleason score remains practically the strongest predictor of the natural course of the disease [7] and is a principal part of nomograms predicting the preoperative stage or post-therapy survival [8,9,10,11,12].

## 3. Contemporary Practices in Gleason Grading–ISUP Grade Groups

In almost 60, years medicine has taken a huge step forward in both clinical and pathological practices. Following the accumulating new knowledge on clinical significance of histological changes, more precise diagnostic tools (such as ultrasound-guided core biopsies and immunohistochemistry refining benign vs. malignant glandular arrangements), and advancements in treatment modalities, the Gleason scale has been updated throughout the years.

The original Gleason system had many drawbacks; back in those days even some of the current histological forms of PCa were not even recognized. Neither was it recommended to report about the existence of minor high-grade components. In addition, there were no recommendations concerning biopsy sampling and the only diagnostic tool available was digital rectal examination. Some of the original criteria of grading, especially in the grade patterns 1 and 2, have been shown to present non-malignant changes or very indolent disease. Thus, currently grade patterns 1 and 2 are not recommended for biopsies at all and grade pattern 2 very seldomly in surgical specimens [13,14].

Based on the above development, the first major changes to the Gleason grading system were introduced by the International Society of Urological Pathology in 2005. During that meeting, the grading system underwent major revision. Medical professionals came to the conclusion that Gleason pattern 3 presenting large cribriform glands should be now graded as Gleason pattern 4. The update to the “modified Gleason” leads to a big discrepancy when analyzing data prior to 2005. However, no consensus was achieved on poorly formed and fused cribriform glands.

Later on, during the 2014 ISUP consensus conference it was accepted that all cribriform structures were to be graded as Gleason Pattern 4. This is due to the fact that numerous studies demonstrated that cribriform glands were related to adverse prognosis regardless of their morphology [15,16,17,18]. Some pathologists, however, had practiced this scheme for some time before the consensus meeting [13]. It was also agreed that Glomeruloid structures should be included into Gleason 4. Mucinous structures were to be graded according to their underlying growth pattern rather than all being marked as Gleason 4. When it comes to intraductal carcinoma of the prostate (IDCP), scientists came to a consensus that only invasive carcinoma should be incorporated into the Gleason score. During the 2014 meeting a new five tier Grade group system was accepted to be reported together with the Gleason score [15].

The most recent ISUP consensus meeting took place in 2019. The major changes to the Gleason grading system concerned the minor/tertiary histological patterns, systematic reports concerning multiple biopsy cores and MR fusion biopsies. It was agreed that minor/tertiary Gleason 4 and 5 grades should be noted in prostatectomies and in biopsies in cases where they constitute >5% of the volume [19].

In 2019, there was another independent working group, Genitourinary Pathology Society (GUPS), that also published their own cancer grading manuscript. ISUP 2019 and GUPS contain many similarities, but some key areas have major disagreements when it comes to recommendations and conclusions. According to GUPS, IDCP should not be included in determining the final Gleason score but it should nevertheless be reported. It is also recommended by GUPS to perform immunohistochemistry in case of ICDP if the Gleason score is 6 or there is a suspicion of cribriform carcinoma. According to ISUP on the other hand, IHC is unnecessary as ICDP should be incorporated automatically in the Gleason score. Another significant disagreement arose around minor/tertiary patterns. According to ISUP, even high-grade components, in the presence of only two histological types of malignant tissue, should be graded as minor when they represent <5%. According to GUPS three different grades have to be present and if grade 5 represents >5%, it should be accounted as a secondary pattern. According to GUPS “minor pattern” is the recommended form with the exception of Grade groups 2 and 3 where “tertiary pattern” should be used [20].

## 4. Quantitative Gleason and Artificial Intelligence

Traditional Gleason grading describes the histopathological state of the prostate in a reasonably rigid categorical fashion which subjects it to big inter-observer variability between growth patterns. The idea behind quantitative Gleason grading is that the cancer grading should rather be treated as a continuum and the categorical grade patterns should be broken down into proportional scales of different grades. According to Sauter et al. 2016 quantitative Gleason grading is less prone to interobserver bias and provides more detailed information than the traditional grading categories [21]. It was also shown that presence of high-grade tertiary patterns have a stronger correlation with negative outcome rather than the area of secondary pattern of equal grade [22].

The use of proportionate grading is becoming easier with better access to image repositories and related data, as well as increased computational ability for complex artificial intelligence (AI) models. Research has shown that the accuracy and consistency of AI algorithms, when it comes to grading, are similar or even better than that of the general pathologists [23,24,25]. AI could make cancer detection and grade assessment more efficient and less prone to human error in the future. Pantanowitz et al. showed high accuracy of image analysis in an AI-based algorithm meant to identify and quantify PCa, recognize low-grade and high-grade tumors and also detect perineural invasion in core needle biopsies [26]. There definitely is a new era of AI-augmented histological image analysis trying to come up with new methods for prognostication as well.

Until now, modifications to the Gleason grade have concerned the histological appearance. With new knowledge and computational technologies available, there is room for further development in PCa grading. Also, with advancement in machine learning and AI technologies, new predictive algorithms could be possible to aid clinicians in their work. We are living at the beginning of a revolution where computers can perform complex cognitive tasks interactively with humans or independently. Big data combined with computer-aided analysis can lead to recognition of complex patterns that were very time consuming or were even never thought of before. This creates a possibility of integrating Gleason grading with advanced genomic studies, biochemistry, multiparametric imaging and electronic patient records to create tools for risk stratification and decision-making in a longer time perspective [24]. Especially, when thinking about active surveillance versus radical treatment, there is a need for more prognostic factors in order to somewhat predict the possible progression of cancer over time [27].

There have been multiple publications describing correlation between a variety of biomarkers and prognosis. Such methods after validation could constitute a whole new tool for the PCa diagnostics and creation of a personalized treatment plan. In the following chapters, we describe the current knowledge on predictive tissue biomarkers at the time of diagnosis for clinically local PCa. We especially concentrate on the clinically relevant end-point of disease-specific survival, overall survival and metastasis-related survival. The independent prognostic significance of the biomarkers is weighted against the established clinical parameters, such as Gleason grading.

## 5. Systematic Review of Biomarkers Related to Clinically Relevant Endpoints

In order to curate publications of prognostic or predictive (which by a strict definition means comparing different treatment groups) biomarkers related to clinically relevant end-points after treatment of local PCa, we conducted a systematic literature search. Please see the PRISMA diagram (Figure 1) for an overview of the systematic review strategy.

A systematic Pubmed query was performed in October 2020 (search parameters are provided in Appendix A) for results between 2010 and 2020. The search returned *n* = 4377 results, of which 4104 remained after removing duplicates. Titles and abstracts were downloaded for all non-duplicated papers using the R (version 4.0.1), package easyPubMed. We then applied regular expression key-word string searches for inclusion criteria terms regarding survival analysis with regards to study-defined “hard outcomes” (metastasis-free survival, overall survival, disease-specific survival), or primary therapy, which resulted in *n* = 1569 articles being taken forward for screening. These were screened for inclusion criteria from the title and abstract, which resulted in *n* = 128 papers being taken to full-text screening. During full text screening, we sought to determine whether biomarkers were assessed in multivariate regression analyses against hard outcomes of metastasis-free survival, overall survival, and disease-specific survival, which resulted in *n* = 55 publications for further reporting. Please refer to Table 1 for the final summary of systematically reviewed articles in our study.

As most of the studies consider RP patients and less frequently radiation therapy (RT) patients, we stratified the biomarkers based on the source of biological specimen and method of analysis. This created groups of tissue biopsy, tissue RP, blood as sources of biomarkers, and connected them to methods of analysis such as immunohistochemistry, PCR, ELISA, FISH, and RNA hybridization.

In order to focus on studies of long term outcomes of primary PCa, we also excluded any reports that analyzed patient cohorts in advanced stages or under adjuvant treatment, including CRPC. Androgen deprivation therapy combined with RT was accepted as it is a standard combinatorial treatment for many of the local PCas. Thus, the review included only patients at the time of diagnosis or primary treatment having hormone naive PCa. Additionally, during the composition of the systematic review part, we curated relevant literature that was missed by the Pubmed search terms but met the same inclusion criteria. These publications were then cited accordingly in each relevant section in the results. 

### 5.1. Results

Table 1 summarizes the finding of the systematic review into distinct categories based on the practical approach concerning the source material and the analysis method.

#### 5.1.1. Biopsy-Based Biomarkers—Radiation Therapy and Radical Prostatectomy

At the time of diagnosis, the treatment decisions are based on cancer grade in biopsies. The standard treatments with radical intent for localized PCa are prostatectomy and radiation therapy. The choice of treatment depends on conventional risk-assessment and patient-related factors, including expected life-years after the diagnosis. For more individualized risk assessment, weighting clinically relevant cancer-related endpoints against competing causes of death is essential. Nomograms built combining conventional risk parameters such as PSA (including PSA-kinetics and density), clinical stage, proportion of cancer in biopsies, and Gleason score are beyond this review, and we are concentrating on putative biological signatures as biomarkers. First, we describe the biomarkers based on a systematic review that have been shown in diagnostic biopsies to predict either metastasis or death during the follow-up after radical treatment.

Certain gene panels assessed in biopsies and that have been commercialized have shown independent significance for clinically relevant events after radical prostatectomy. Prolaris, a 31-gene cell cycle progression panel assessed in biopsies before RP can predict metastasis formation after the treatment [82]. Decipher, a commercially available 22-gene RNA transcript-based algorithm is predictive of metastasis after primary treatment whether it is RT or RP together with adjuvant androgen deprivation therapy (ADT) [62,83].

Another commercially available gene panel is Oncotype DX. A validated and commercially available biopsy-based 17-gene panel relying on polymerase chain reaction assay was tested by Van Den Eeden et al. Almost 300 men were treated with RP only and a Genomic Prostate Score (GPS—scale 0–100) was assessed. This panel turned out to be independently significant for MFS and DSS. The higher the score the worse the outcome. According to the authors patients with low scores did not present with metastases during the observational period [63].

In general, proliferation is considered low in primary PCa. However, common proliferation marker Ki-67 has shown predictive value in a number of studies. As a stepwise variable it showed independent significance for poor outcome in terms of MFS and DSS by Tollefson et al. Ki-67 was included in their so-called Mayo model together with Perineural Invasion, and Gleason Score. This biopsy-based panel was compared with the RP outcome. As a conclusion the authors suggest that Ki-67 should be routinely assessed in biopsies together with perineural invasion and Gleason [28]. Verhoven et al. also showed independent significance for the same endpoints (MFS and DSS) as in the abovementioned study with dichotomized Ki-67 using a 6.2% cut-off for a group of patients treated with RT ± ADT [29]. The differences between RT + ADT and RT alone arms were measured only in univariate analysis [29]. Ki-67 has also been shown to be independently significant for poor MFS by Pollack et al. in a study of four different genes. Ki-67, MDM2 and Cox-2 correlated positively with poor outcome whereas p16 had a negative correlation. In other words, high expression of p16 was protective since it is a tumor suppressor gene. All are independently significant predictive factors among patients treated with RT and either long or short term ADT. The authors suggest that they should be used together as a gene panel. They even created a hypothetical 65-year-old patient model with fixed values of PSA, Gleason and T scores. This patient’s characteristics showed a difference in 10-year predicted risk when compared against favorable and unfavorable biomarker status [30].

Neuroendocrine (NE) characteristics are usually acquired during ADT, and primary NE carcinomas are rare. Increased expression of NE biomarker Chromogranin as a continuous variable was independently significant for shorter MFS and DSS in patients treated with RT ± ADT. Krauss et al. admit that most of the patients selected for this analysis had a Gleason score 8–10 which may have led to a selection bias of more patients receiving ADT [31].

Periostin is a secreted extracellular matrix protein that interacts for example with integrins and might promote cancer epithelial mesenchymal transition [84]. High Periostin expression is an independently significant predictor of poor OS as well as a composite outcome including radiographic progression and recurrence-free survival. The study by Cattrini et al. was performed on patients treated with RP, RT, or ADT for locally advanced/metastatic disease [32].

TMPRSS2-ERG is one of the first fusion genes detected in solid human tumors [85] and it is present in approximately 50% of localized PCas. A study by Zeng et al. showed independent significance of TMPRSS2-ERG fluorescence in situ status as a prognostic factor for poor OS in patients treated with RP. Material for the purpose of this study was provided by both RP and biopsies [64]. ERG immunohistochemistry has been shown to correlate strongly with the fusion-gene status but the prognostic value for MFS, DSS, or OS in biopsies does not seem to be significant. We discuss later its role together with tumor-suppressor PTEN in survival prediction.

Loss of tumor-suppressor Disabled homolog 2-interacting protein (DAB2IP) is an independently significant predictor of poor MFS on the basis of pretreatment biopsy among patients treated with RT and ADT according to Jacobs et al. They also suggest that DAB2IP loss might explain the differences in tumor aggressiveness and radiation resistance. In the same study the expression of EZH2 was also measured for hard endpoints but statistical significance was not observed since almost all PCa samples expressed this biomarker. The authors suggest that higher expression of EZH2 could be used as a screening for high-risk patients since this biomarker was observed in poor outcome patients [33]. Another, and one of the most studied, tumor-suppressor genes is phosphatase and tensin homolog on chromosome 10 (PTEN) which is lost in approximate 20% of PCas. The loss of PTEN releases PI3K pathway activated by growth factors and leads to, e.g., AKT and mTOR-regulated cellular growth, survival and cancer cell migration. Its various roles in cancer are beyond this review but we will focus on its prognostic role in human samples.

DNA repair-related genes have been shown to be altered and inactivated in PCa as well. Castro et al. analyzed the relevance of BRCA1 and 2 mutations in two separate cohorts. One was a biopsy-based RT cohort and in the second patient were treated with RP. In both cohorts, some patients received ADT, however it was much more common in the RT cohort. Since there was no significant statistical interaction between BRCA status and treatment modality for either of the two endpoints, separate prognostic analyses were not justified according to the authors. An assumption was made that BRCA affected both cohorts similarly. Finally, BRCA carriers were proven to have worse DFS and MFS [65]. Na et al. found similar results, in a cohort of primary PCa patients, and identified that patients harboring a BRCA mutation had significantly poorer DSS [86].

#### 5.1.2. RP Specimen-Based Markers

Decipher, a gene panel mentioned above, is independently significant in predicting disease specific survival as proven by Cooperberg et al. In this study patients were mainly treated with RP but also with or without adjuvant RT and adjuvant ADT [66]. Ross et al. concluded that Decipher is correlated with MFS [67]. Both of these papers concentrated on post prostatectomy risk assessment.

Liu et al. used Affymetrix 6.0 single-nucleotide polymorphism microarrays to screen chromosomal copy number alterations (CNAs). Among all the CNAs tested only two genes showed independent significance in MVA. PTEN loss and MYC gain were either separately or jointly prognostic factors of poor DSS. The main cohort consisted of patients treated with RP only and the abovementioned results were confirmed with three other cohorts (MSKCC, KUH, JHH) [87]. Zhao et al. used an Affymetrix GeneChip array to analyze RNA transcript measures in four prostatectomy cohorts (including one in which patients received RT after surgery). High expression of RNA transcripts in five genes were identified as being prognostic. Three or more of the following proteasome genes (PSMB4, PSMB7, PSMD14, PSMB2, and PSMD11) have to be present in order to be independently significant for poor MFS. The authors also found that the correlation was more pronounced among younger patients suggesting that this biomarker could be used in this specific age group and also puts the use of proteasomal inhibitors among older groups under a question mark [68].

Zhao et al. initially tested 20 genes in patient cohorts treated with RP and in some cases with adjuvant radiotherapy. Out of these, only the three showed independent significance and a novel outliner gene panel was created. The outlier score from the triple gene panel including NVL, SMC4, and SQLE was independently associated with poor MFS and OS. The three top outlier genes were later validated [69].

Androgens are shown to have potential to regulate the expression of protein kinase A (PKA) and its subunits in PCa cells. The PKA pathway is also linked to the androgen receptors in PCa. Moen et al. showed that low expression of Cβ2 subunit of PKA by mRNA analysis in PCa samples was an independently associated prognostic factor of poor DSS. The study was divided into three different cohorts where patients were mostly treated with RP but also with or without RT/ADT. RNA analysis of Cβ2 was dichotomized with a cut-off value of 130 [70].

Evans et al. developed a novel patient-level gene set enrichment analysis-based pathway profiling approach that they further applied on patients treated with RP with or without adjuvant RT/ADT. The authors used Affymetrix Human Exon 1.0ST GeneChips, which included a panel of 17 DNA damage and repair (DDR) pathway genes. Over 1000 patients were divided into a training cohort and three validation cohorts. The DDR signature was proven to be an independently significant prognostic factor associated with poor MFS and OS among younger patients (<70). This biomarker could be used as a prognostic tool in risk stratification, according to the authors. Correlations with AR and ERG expression were also analyzed and were proven to be significant [71]. Another DDR-related work by Castro et al., which was mentioned above, showed that not only in pre-treatment biopsies but also in RP specimens, BRCA1 and 2 gene mutations are independently significantly prognostic for poor DFS and MFS [65].

Gene expression analysis from RP specimens by Hu et al. showed that low AXIN2, a protein involved in negative regulation of beta-catenin and wnt pathway, expression in RT-PCR was proven to be independently significant for poor MFS. In this cohort, patients were treated primarily with RP but also in some cases with adjuvant RT and neo/adjuvant ADT. The prognostic significance of AXIN2 expression was further externally validated in a separate cohort. The study even showed in vitro and in vivo correlation of low AXIN2 levels with increased invasiveness and tumor growth. As a conclusion, the authors suggest that this gene might be used in creation of targeted therapy as it is strongly associated with neoplastic growth [72].

MicroRNAs (miRNAs) are involved in various key cellular processes including proliferation and differentiation. They are also diverse epigenetic regulators of malignant transformation [88]. Schmidt et al. trained and validated a novel 4-miRNA prognostic ratio model meant to stratify the risk of post RP patients. Six potential prognostic miRNAs were initially identified, leading to the creation of a 4-miRNA model (*MiCaP*). The authors assessed a ratio of 4 mi-RNAs (miR-10b-5p, miR-23a-3p, miR-133a, and miR-374b-5p), and found that that the combined MiCaP panel when analyzed in RP specimens was prognostic of DSS. The presented data was later validated in two independent cohorts [73]. Low miR-424-3p expression among patients treated with RP is a significant independent factor of poor composite outcome (clinically palpable tumor or metastasis verified by radiology) as shown by Richardsen et al. Biomarker expression was dichotomized into low vs high. According to the authors, the abovementioned results highlight the importance of miR-424-3p being a potential target of therapeutic treatment [74]. Another short non-coding RNA was presented by Laursen et al. in a study based on patients treated with RP only. High expression of miR-615-3p appeared to be a prognostic factor of poor DSS. The material was analyzed using miRNA-PCR and the data acquired was dichotomized [75].

Elevated protein expression of estrogen receptor ER(α) and low ER(β) are both independently significant prognostic factors of poor OS and composite outcome (local recurrence and distant metastasis) among patients treated with RP, according to Megas et al. [35]. In a study performed on patients treated with RP only by Grindstad et al., in addition to the two above-mentioned biomarkers, the significance of aromatase was also analyzed. In contrast to the results reported by Megas et al., the authors reported that higher ERα was associated with better DSS and a composite outcome including local symptomatic recurrence and/or findings of metastasis to bone, visceral organs, or lymph nodes. The authors also found that higher aromatase expression was associated with better composite outcome. The combination of these two biomarkers was independently significant for better DSS and composite outcome. ER(β) did not show independent significance [36].

In terms of hormonal regulation markers, a combined IHC analysis of steroid and xenobiotic receptor (SRX) and its target gene cytochrome CYP3A4 was performed by Fujimura et al. among patients treated with RP and, in some cases with adjuvant androgen deprivation and/or radiation therapy. Lowered expression of both of these biomarkers was shown to be independently significant prognostic factors of worse DSS. The authors suggest that the fact, that SRXr regulates the cytochrome-mediated inactivation of testosterone, could be used as a target for therapy in PCa [37].

Cell cycle regulator p53 nuclear protein expression reflects mutation in *TP53 gene* and has recently shown to relate with poor MFS and DSS independently among patients treated primarily with RP ± adjuvant RT/ADT. Quinn et al. [38]. A second gene that is directly associated with cell cycle regulation is protein phosphatase magnesium-dependent 1 delta (PPM1D), which has been shown to inhibit p53 function during oncogenesis. According to Jiao et al., higher expression of this biomarker among patients treated with RP is significantly correlated with poor OS [39]. 

Transformer 2β (Tra2β), a splicing regulator-related protein was studied by Diao et al. with IHC in RP cohort of 160 patients. Tra2β expression was dichotomized into high and low expression groups. The former one was independently associated with poor OS. The authors suggest that Tra2β could be used as a target of novel therapy [40].

Microtubule-associated protein 1 light chain 3B (LC3b), a component of autophagosome formation, was analyzed by Mortezavi et al. in patients treated with RP ± neoadjuvant ADT. This biomarker was shown to be independently associated with better DSS [41]. BAG family molecular chaperone regulator 3 (BAG3), on the other hand, is a co-chaperone protein related to autophagy. Its intra-cytoplasmic delocalization is a specific feature of neoplastic transformation. This feature correlated with poor MFS among patients treated with RP only, according to Staibano et al. According to the authors, this pathway could play a key role in overcoming chemoresistance of cancer cells [42]. Positivity for Hairy/enhancer-of-split related with YRPW motif protein 2 (Hey-2), a transcription factor, predicted increased risk of distant metastasis by 5.6-fold compared to negative staining among patients treated with RP only, according to Tradonsky et al. [43].

A multi-institutional study showed that elevated nuclear frequency of p65 (a NF-κB subunit), as measured by IHC, is an independently significant prognostic factor of development of bone metastasis and PCa-related death after RP [44]. Different subunits of NF-κB function with various mechanisms regulating tumor initiation and growth, one of which is immunological environment regulation. Programmed cell death protein (PD-1) is a T cell suppressive immune checkpoint molecule that has been shown to have a role in many cancers. High density of PD-1 expressing intratumoral stromal lymphocytes was shown to predict quicker local recurrence and distant metastasis among patients treated with RP only in a study by Ness et al. [45]. Although therapies targeting PD-1 and PD-L1 have not been successful in PCa, more detailed prognostic information might lead future clinical trials in selected patients.

High Neprilysin (or CD10) expression in local lymph node metastasis is predictive of poor OS among patients treated with RP ± ADT according to Fleischmann et al. [46]. This might be related to the function of CD10 as a modulator of tumor microenvironment during carcinogenesis. Another extra-cellular matrix modifying enzyme as biomarker was presented by Nonsrijun et al. Overexpression of matrix metalloproteinase 11 (MMP-11) in RP specimen showed independent significance as a prognostic factor of poor DSS [47].

Among patients treated primarily with RP but in some cases with adjuvant RT and ADT, loss of PTEN expression is an independently significant prognostic factor of poor OS and MFS, according to Hamid et al. The authors studied PTEN with quantitative fluorescence IHC as continuous and dichotomized variable [48]. Lahdensuo et al. showed similar results; however, here, ERG negativity paired with PTEN loss was a predictive factor of poor DSS. Dichotomized IHC expressions of biomarkers were measured in patients treated with RP only [49]. In a separate study of ERG status and obesity, the authors found that increased BMI and waist circumference, particularly in ERG positive patients, was significantly associated with poor survival [89]. Troyer et al., also studied PTEN status, as assessed by FISH, in RP specimens. The authors found that PTEN loss in RP was associated with poor composite outcome including DSS, MFS, clinical recurrence and salvage treatment. According to the authors, the FISH method is very promising as it requires less specimen material and could be applied in diagnostic biopsies [76]. 

Myosin phosphatase target subunit 1 (MYPT1) is inhibited by miR-30d in PCa, in a pro-angiogenic and tumorigenic pathway. Low immunoreactivity score of MYPT1 correlates with poor OS in RP patients according to a study by Lin et al. [50]. Nordby et al. studied another pathway that could be a target of novel anti-angiogenic treatments in the future. In their study, VEGFR-2 expression was associated with poor composite outcome including local recurrence and distant metastasis among patients treated with RP without any form of adjuvant therapy [51]. Neuropilin-2 (NRP2) is a transmembrane protein that acts as a coreceptor of VEGF and promotes tumorigenesis possibly through mTORC2 as well [52,90]. Elevated expression of NRP2 protein (dichotomized values) is an independently significant prognostic factor of shorter DSS among patients treated with RP, according to Borkowetz et al. Stratification by risk factors made the prognostic value more pronounced with high-risk PCa [52]. Another study that researched the role of angiogenesis in PCa was published by Liu et al. Here, the presence of vasculogenic mimicry was also related to adverse OS and a composite outcome consisting of local recurrence and distant metastasis [53].

Platelet derived growth factors (PDGFs) and their receptors (PDGFRs) are known to be significant regulators of mesenchymal cells in different types of cancers. Nordby et al. showed that patients treated with RP, and in some cases with postoperative ADT or RT, had more frequently local recurrence and distant metastasis in case of high expressions of PDGFR-β [54]. In this study, the IHC scoring was dichotomized into low and high expressions. The function of Pleomorphic adenoma gene like-2 (PLAGL2) is not precisely known. Research has shown however that it is a transcription factor related to tumorigenesis. According to Guo et al. (PLAGL2) overexpression is associated with poor OS among patients treated with RP only [55].

Golgi phosphoprotein 3 (GOLPH3) is an oncogene that has been shown to be involved with tumorigenesis in PCa and other neoplasms, and regulates (along with other trans-Golgi matrix family proteins) rapamycin signaling [91]. According to Zhang et al., GOLPH3 positivity is correlated with reduced OS in patients treated with RP only. Data was dichotomized for this MVA but authors suggest that a numeric intensity score could be used for measurement of the aggressiveness [56].

Tretiakova et al. performed a multi-institutional study of Ki-67 expression in 1004 prostatectomy specimens. The authors found that high Ki-67 proliferation index (measured as a continuous variable) is a prognostic factor of poor OS, DSS, and MFS (as part of a composite outcome additionally measuring clinical recurrence, treatment with salvage therapy, and prostate-cancer specific mortality). They suggest that Ki-67 has potential to be measured during routine biopsies for patients during AS [57]. On the other hand, Pascale et al. analyzed the statistical significance of Ki-67, neuron specific enolase (NSE), chromogranin A (ChrA), and synaptophysin (Syp) among patients treated with RP and transurethral resection of the prostate (TURP). Regardless of the treatment, only Ki-67 was an independently significant prognostic factor of poor OS [92].

The solute carrier family 18 member 2 (*SLC18A2*) is responsible for encoding genes related to transmembrane vesicular transportation. Loss of this gene is associated with poor OS, according to Haldrup et al. The authors suggest that the survival analysis should be repeated with a longer follow-up time [58].

Phosphoinositide 3-kinase (PI3K)/Akt pathway is usually activated by PTEN loss. Recent studies have however shown that loss of INPP4B has similar effects [93]. Rynkiwicz et al. performed a cohort study on patients treated primarily with RP but in some cases also with adjuvant RT or ADT. INPP4B loss was correlated with more frequent local recurrence and distant metastasis [59].

High MUC1 expression in lymph node metastasis is independently significant for unfavourable outcome in terms of DSS, according to Genitsch et al. in a study on patients treated with RP only [60]. Interestingly, it has been previously shown that MUC1 downregulates the androgen receptor at the transcriptional level in cell lines [94], which may indicate that MUC1 could be amenable for targeted therapy.

#### 5.1.3. Peripheral Blood Markers

Increased plasma fibrinogen level is a significant independent predictor of poor DSS and OS in patients treated with RT (±adjuvant or neoadjuvant hormonal therapy), although the clinical parameters were stratified into binomial low and high-risk categories in the report by Thurner et al. [77].

Leukocyte relative telomere (RTL) length shows independent significant correlation with worse DSS and OS in patients treated with RT (±adjuvant or neoadjuvant hormonal therapy) in a report by Renner et al. [78]. Patients were stratified into low-, intermediate-, and high-risk groups according to the NCCN guidelines. 

Steroidogenic germline polymorphism of certain SNPs (CYP1B1 (rs1800440), COMT (rs16982844), and SULT2B1 (rs12460535, rs2665582, rs10426628)) was shown to be independently significant for negative outcome in OS and a composite outcome (including resistance to androgen-deprivation therapy, metastasis, and/or death) by Lévesque et al. in a cohort of RP patients. Patients were stratified according to risk into four prognostic subgroups. However, out of the tested genes, *SULT2B1* rs2665582 and rs10426628 are protective. The study suggested that combination of these markers, rather than each SNP individually, should be used for better outcome prediction Adding a previously discovered *HSD17B2* (rs4243229, rs1364287, rs2955162, rs1119933) to the new biomarkers compared in MVA creates a remarkable 8 gene panel [79]. Single nucleotide polymorphism in Ribonuclease L (RNASEL) was studied in terms of outcome correlations by Schoenfeld et al. A prospective study of patients treated with either RT or RP was performed, presence of rs12757998 gene allele has shown independent significance in the case of the RP cohort. In the multivariate analysis the rs12757998 variant allele was associated with a better composite outcome including both DSS and MFS. Additionally, men homozygous for the abovementioned allele variant were associated with a significantly reduced hazard for the composite outcome when RT was combined with ADT [80]. 

Oncofetal protein insulin-like growth factor 2 (IGF2) messenger RNA-binding protein 3 (IMP3) was studied among patients treated with RP or palliative TURP by Szarvas et al. High IMP3 serum levels (as a continuous variable) have been shown to be an independent risk factor of DSS in both pre and postoperative models [81]. Szarvas et al. also suggested that in addition to being a biomarker for PCa, IMP3 has potential as an actionable immunization target

#### 5.1.4. Active Surveillance

Active surveillance (AS) is a standard of care option for primary treatment of low to even some intermediate risk PCa [95], and prognostic biomarkers of AS outcomes is an area of active research [96]. Given that the aim of this review was to profile prognostic biomarkers of primary PCa, we included AS studies in our initial screen. We did not identify any reports from AS studies with hard outcomes in our systematic literature search. This is not entirely unexpected, given that AS has only become a viable treatment option within the past 10–15 years, and most studies assessing outcomes of AS analyze easily attainable, short-term surrogate outcomes such as grade elevation in subsequent biopsies or change to active treatment. Our search terms did not contain watchful waiting (WW) or deferred treatment, which can include higher-risk primary PCa patients, whose risk-profiles differ from those undergoing contemporary AS. Despite this, we identified two relevant studies reporting associations of biomarkers assessed in prostate biopsies and TURP specimens, and hard outcomes.

Hammarsten et al. reported that high stromal Caveolin-1, a scaffold protein involved, for example, in signal transduction and interacting with integrins, RNA expression is associated with favorable disease-specific survival in men under WW [61]. Kammerer-Jacquet et al., reported results that a linear increase in prostate biopsy Ki-67 protein expression, as detected by IHC, predicts elevated risk of disease-specific mortality, especially in GG1&2 PCas [34]. Despite inherent difficulty in interpreting these results in light of the current standard of care, these results suggest a biological correlation of these markers with outcome that is independent of grade (and arguably treatment). The findings from Ki-67 protein expression are in line with previously discussed results [28,29,30,57], that consistently report a positive association between increased expression and poorer outcome. Indeed, Hammarsten et al. further went on to report a similar association between Ki-67 and poor outcome in the same WW cohort, however, they did not analyze the prognostic value of Ki-67 with other clinical variables in a multivariable model [97]. Similarly, other recent reports assessing Ki-67 expression status in prostate biopsies have also reported independent association with DSS [98,99]. Reports from other WW studies have reported an independent association of biomarkers and WW outcomes [100,101].

A number of commercial gene panels have been proposed for use in prognostication of AS. Prolaris has been shown to be prognostic of PCa outcomes in prostate needle biopsies [102]. Other commercially available gene panels such as Decipher [103,104] and Oncotype Dx [105] have also been assessed in AS cohorts. However, these studies focused on associating biomarker status with AS inclusion criteria, or short term AS outcomes such as grade progression as subsequent biopsy, or adverse findings in pathology in RP specimens. Further work will need to be done to assess these biomarkers identified in this section in the context of other currently available commercial gene panels, modern clinical standards of care such as in mpMRI targeted biopsies, and long term follow up in AS cohorts.

## 6. Discussion

The aim of our systematic review was to find putative prognostic biomarkers with strict criteria for the follow-up end points, i.e., metastasis formation, PCa-specific death and death from any cause. Studies with BCR only during the follow-up were categorically excluded. Another important criterion for biomarker inclusion in our review was independent prognostic value when adjusted for the clinical conventional risk-defining parameters. Only in this way can one expect to find markers with potential to fill the unmet need for higher-resolution risk stratification and personalized approaches.

The biomarker review revealed that there indeed are significant findings on markers that relate to clinically relevant follow-up events and are independent of clinical characteristics. At the same time, it revealed the heterogeneity in methodology and reporting of these studies. Markers are very often adjusted to clinical variables or categorized in a way that would not necessarily be applicable in clinical practice but serves the purpose of an individual study and the cohort selected. Many of the markers are semiquantitative which creates a problem of reproducibility. Studies mainly lack validation cohorts or the findings have not been reproducible in another cohort later. However, some of the markers may be considered at least partially useful or promising, although the consensus recommendations have found them to be not clinically applicable [106,107,108].

There are several reasons for non-existing clinical predictive markers. Firstly, PCa is a heterogeneous disease and some of the issues finding new stratifying predictive markers may simply relate to, e.g., sampling the relevant lesions and detecting signals correctly in the blood. Strategies using targeted biopsies and well-designed RP TMAs that mimic the biopsy situation may offer solutions for this. Secondly, methodologies vary considerably and only a small number of markers have been validated with methods that are consistent in samples across different laboratories. In the end, a lack of prospective studies comparing different treatment modalities that would include biomarker testing stems from the complexity of designing such a study. Complexities are in patient cohort selection (same stage, standardized treatment modalities) and in required long intervals between initial treatment and clinically relevant endpoints. Additionally, solely a biomarker study would not perhaps attract enough funding if not coupled with hypotheses of finding druggable targets or theranostic molecules. In search of aids for decision-making and individualized prediction, more standardized approaches and larger prospective multicenter studies are warranted.

One approach that our systematic search did not conceptually include is whether a biomarker has additional (although correlative) value in prediction beyond what the clinicopathological parameters inform us. Such statistics include ROC-AUC (receiver operating characteristics—area under curve) analysis by which single parameters or combinations of characteristics can be compared for any addition in predictive significance. This was not the topic of our current review per se but to mention there are e.g., gene expression panel-based and MRI-based studies showing improved predictions on top of clinical parameters for metastatic and lethal disease [109,110,111,112,113]. The future of PCa prognostic work-up is built on multimodal approach, i.e., traditional histology (aided with image analysis and AI solutions), biomarkers, clinical data, and imaging. Although being subjective, Gleason grading has remained the prognostic method with most impact on treatment choices. The divided views on certain approaches in reporting are in the end something that is less important and emphasis should be put into lessening the variation in grade pattern assessment. This all requires local pathology meetings and being able to consult a colleague with a low threshold. Equally important to interobserver discussions is cross-talk with clinicians. There should be local agreement about the reporting of pathology findings as systematically as possible and communication about the pathologists’ approach as it comes to ISUP or GUPS recommendations. Overall, systematic reporting of cancer extent in biopsies facilitates more unified morphology across observers as pathologists need to give more thought to subgrading and proportions of different Gleason patterns. In the systematic approach and in decreasing inter-observer variation, AI will guide subpathology reporting and facilitate unified global recommendations as knowledge accumulates. In the near future, AI may enable the implementation of quantitative Gleason scoring through continuous readouts, providing increased clinicopathological information as compared to traditional stepwise grading.

## 7. Conclusions

Histological grading of PCa remains the gold standard in risk stratification but is revised with quantitative approaches and image analysis algorithms. Latest research has shown that current image based analysis performs as well as visual human Gleason grading in diagnostic workup. There continues to exist an unmet need for biomarkers to further define the prognosis and aid in treatment selection of a localized PCa. Putative biomarkers have been identified but need to be further studied in standardized fashion, most probably complemented with other modalities such as medical imaging.

When it comes to multimodal diagnostic assessment, combining clinical data, histology and imaging is already a recommended approach. In the future, the use of AI and image analysis could provide more personalized approaches to treatment decision-making. e.g., patient’s comorbidities, expected life years and calculated benefit from a particular treatment choice could be amended with biomarker (histology, genomics, or radiomics) based risk-assessment (See Figure 2).

## Figures and Tables

**Figure 1 cancers-13-00628-f001:**
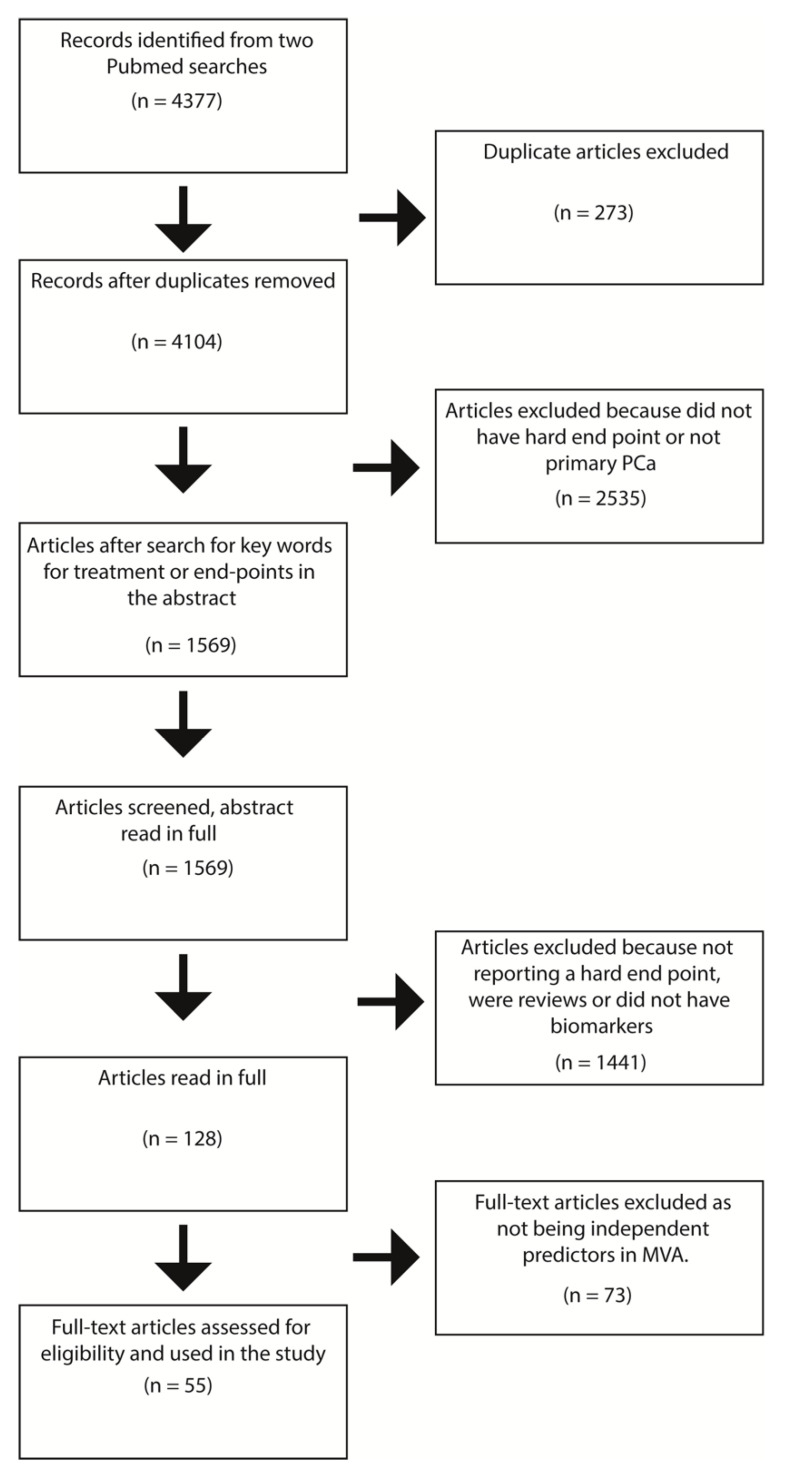
PRISMA diagram of the systematic review. MVA = Multivariable analysis.

**Figure 2 cancers-13-00628-f002:**
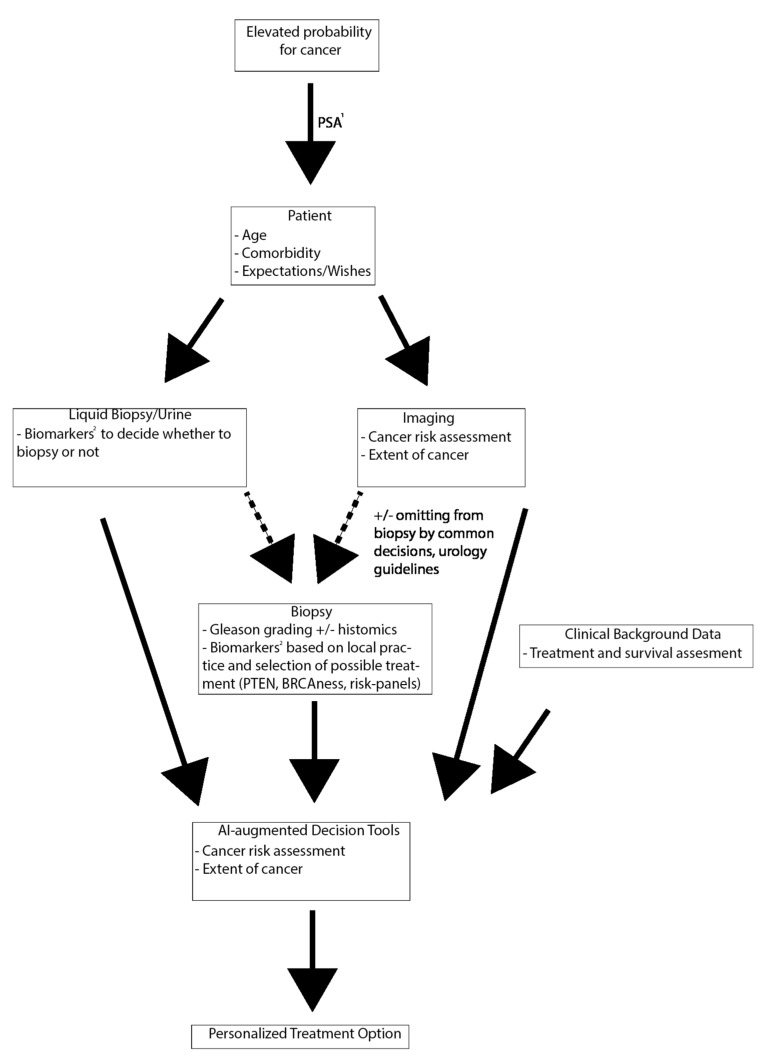
Future prospect of a multimodal diagnostic work-up and treatment decision-making. ^1^: Prostate-specific antigen (PSA) is unspecific at the diagnostic phase and should be taken only as part of an informed risk assessment or based on recommendations for personalized screening. ^2^: Current evidence does not support any biomarkers. Further research is needed to implement biomarkers to clinical practice (as discussed in text).

**Table 1 cancers-13-00628-t001:** Results of the systematic Pubmed search (for detailed inclusion criteria, see text and Appendix A). Biomarkers independently related to clinically relevant outcomes in localized prostate cancer.

Source Material, Analysis	Ref. No.	First Author	Year	Primary Therapy	Biomarkers	Additional Analysis Info	Outcome	Correlation
Biopsy, IHC	[28]	Tollefson	2014	RP	Ki-67		MFS, DSS	Positive
	[29]	Verhoven	2013	RT + ADT	Ki-67		DSS, MFS	Positive
	[30]	Pollack	2015	RT + ADT	Ki-67, MDM2, p16, Cox-2		MFS	Negative/Positive
	[31]	Krauss	2011	RT + ADT	Chromogranin A (CgA)		MFS, DSS	Positive
	[32]	Cattrini	2019	RP, RT, ADT	POSTN		OS, MFS *	Positive
	[33]	Jacobs	2016	RT + ADT	EZH2		MFS	Negative
	[34]	Kammerer-Jacquet	2020	AS	Ki-67		DSS	Positive
RP, IHC	[35]	Megas	2016	RP, RT, ADT	ER(α), ER(β)		OS, MFS *	Positive/Negative
	[36]	Grindstad	2018	RP	ER(α), Aromatase		MFS *, DSS	Negative
	[37]	Fujimura	2012	RP, RT, ADT	SXR, CYP3A4	RP and Western blot	DSS	Negative
	[38]	Quinn	2020	RP, RT, ADT	p53		MFS, DSS	Positive
	[39]	Jiao	2014	RP	PPM1D		OS	Positive
	[40]	Diao	2016	RP	Tra2β		OS	Positive
	[41]	Mortezavi	2018	RP, ADT	LC3b		DSS	Negative
	[42]	Staibano	2010	RP	BAG3		MFS	Positive
	[43]	Tradonsky	2012	RP	Hey2		MFS	Positive
	[44]	Grosset	2019	RP	NF-κB p65		MFS, DSS	Positive
	[45]	Ness	2018	RP	PD-1+ stromal lymphocytes		MFS *	Positive
	[46]	Fleischmann	2011	RP + ADT	CD10	LN+ patients only	OS	Positive
	[47]	Nonsrijun	2015	RP	MMP-11		DSS	Positive
	[48]	Hamid	2020	RP, RT, ADT	PTEN		OS and MFS **	Negative
	[49]	Lahdensuo	2018	RP	ERG, PTEN,		DSS	Negative
	[50]	Lin	2017	RP	MYPT1		OS	Negative
	[51]	Nordby	2015	RP	VEGFR-2		MFS *	Positive
	[52]	Borkowetz	2020	RP	NRP2		DSS	Positive
	[53]	Liu	2012	RP	Vasculogenic mimicry (VM)		OS, MFS *	Positive
	[54]	Nordby	2018	RP, ADT, RT	PDGFR-β		MFS *	Positive
	[55]	Guo	2017	RP	PLAGL2		OS	Positive
	[56]	Zhang	2016	RP	GOLPH3		OS	Positive
	[57]	Tretiakova	2017	RP	Ki67		OS, DSS, MFS *	Negative
	[58]	Haldrup	2017	RP	SLC18A2		OS	Negative
	[59]	Rynkiewicz	2015	RP, RT, ADT	INPP4B		MFS *	Negative
	[60]	Genitsch	2017	RP	MUC1	RP and LN Mets	DSS	Positive
	[61]	Hammarsten	2017	AS + TURP	Caveolin-1	RP and TURP	DSS	Negative
Tissue—other	[62]	Nguyen	2018	RP, RT + ADT	Decipher	exon microarray, Bx	MFS	Positive
	[63]	Van Den Eden	2018	RP	Oncotype DX	RNA-PCR, Bx	MFS, DSS	Positive
	[64]	Zeng	2016	RP	TMPRSS2-ERG	RP and Bx, FISH	OS	Positive
	[65]	Castro	2016	RP/RT + ADT	BRCA1 and 2	Mutational analysis, Bx and RP	DSS, MFS	Positive
	[66]	Cooperberg	2015	RP, RT + ADT	Decipher	RNA hybridisation, RP	DSS	Positive
	[67]	Ross	2016	RP	Decipher	RNA hybridisation, RP	MFS	Positive
	[68]	Zhao	2016	RP, RT	PSMB4, PSMB7, PSMD14, PSMB2, PSMD11	RNA microarray hybridization, RP	MFS	Positive
	[69]	Zhao	2016	RP	NVL, SMC4, SQLE	qRT-PCR	MFS, OS	Negative
	[70]	Moen	2018	RP, RT, ADT	catalytic subunit Cβ2	RNA nanostring, Bx	DSS	Positive
	[71]	Evans	2016	RP, RT, ADT	17 genes, DDR pathway	GSEA, RP	OS, MFS	Positive
	[72]	Hu	2016	RP, RT, ADT	AXIN2	qRT-PCR, RP	MFS	Negative
	[73]	Schmidt	2019	RP	4-miRNA ratio model (MiCaP)	miRNA PCR, RP	DSS	Negative/Positive
	[74]	Richardsen	2020	RP	miR-424-3p	miRNA ISH, RP	MFS *	Negative
	[75]	Laursen	2020	RP	miR-615-3p	miRNA-PCR, RP	DSS	Positive
	[76]	Troyer	2015	RP	PTEN	FISH, RP	DSS *	Negative
	[53]	Liu	2013	RP	PTEN, MYC	SNP array analysis, RP	DSS	Negative/Positive
Blood	[77]	Thurner	2016	RT + ADT	Plasma fibrinogen level	Fibrinogen assay	DSS, OS	Positive
	[78]	Renner	2019	RT + ADT	Leukocyte relative telomere (RTL)	DNA-PCR	OS, DSS	Positive
	[79]	Lévesque	2015	RP	CYP1B1, COMT, and SULT2B1 (3 SNPs)	SNP genotyping	OS, MFS *	Positive/Negative
	[80]	Schoenfeld	2013	RP, RT + ADT	Ribunuclease-L (rs12757998)	SNP genotyping	DSS and MFS **	Negative
	[81]	Szarvas	2014	RP + TURP	IMP3	ELISA	DSS	Positive

AS = Active surveillance, Bx = Biopsy, RP = Radical prostatectomy, RT = Radiotherapy, ADT = Androgen deprivation therapy, IHC = Immunohistochemistry, ISH = In situ hybridization, PCR = Polymerase chain reaction, GSEA = Gene set enrichment analysis, TURP = Transurethral resection of the prostate, DSS = Disease-specific survival, OS = Overall survival, MFS = Metastasis-free survival. *: Outcome is a composite outcome including non-hard outcomes. **: The study did not analyze the outcomes separately.

## Data Availability

Not applicable.

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
