# Peer review of "Grading Evolution and Contemporary Prognostic Biomarkers of Clinically Significant Prostate Cancer"

_cancers, 2021, doi:10.3390/cancers13040628_

Round 1

Reviewer 1 Report

The article ‘Grading evolution and contemporary prognostic biomarkers of clinically significant prostate cancer’ by Konrad Sopyllo et al., is highlighting the improvement of pathological grading of prostate cancer over time and its limitation in certain cases. The article is also extensively reviewing the so-called biomarkers for determining the advancement of primary prostate cancer and also nicely justify how they disconnected with the pathological grading and treatment stratification. Overall this review article comprehensively highlighting the present scenario of prostate cancer diagnosis, classification, and treatment management. Based on the importance of the content, I have a favorable recommendation for this article.

Author Response

Thank you for the favorable comments. 

Reviewer 2 Report

This is a well written review and captures the contemporary issues around Gleason grading and prognosis.The systematic review is good.

Overall it is well organised and an easy reference document for someone beginning in the field; for  those who are more familiar it may be less useful.

However the discussion and conclusion are limited.

I realise it may be difficult, but the authors might consider is how to use this review to clearly articulate current and future challenges for the field and for pathologists.

To mention a few: a clear agreement and  consensus does not seem to be clear, and there is an impression that discrepant views have precluded universal agreement on how to use Gleason grading, i.e. pattern recongition to classify prostate cancer. How important is subpathology reporting and what is needed to get consensus so that there are clear recommendations for pathology/urology practice worldwide. How would we combine the use of pathology with biomarkers and data from new imaging techniques?

Author Response

We thank you for your constructive comments! We have used the opportunity to revise and amend the text accordingly and hope that our message finds the reader better. We have clarified the somewhat incorrect impression that Gleason grading would have substantial problems in terms of interobserver consistency. The Gleason score would not hold its place as probably the most important prognostic factor if there would not be agreement on the most important aspects of pattern recognition. However, we have emphasized the importance of continuous cross-talk between the pathologist and the importance of systematic reporting. This also lessens the variation in subpathology (not only morphology) reporting which is important especially in Grade groups 2 and 3.

The current and future challenges are highlighted more now, and we have also added a flow chart about the possible future multimodal approach as we hopefully gain more knowledge on biomarkers and artificial intelligence augments the diagnostic work-up and treatment decision-making. This need for revision was also addressed by the editors.

Please see revised version where the changes to Discussion and Conclusions are visible. We have also added Figure 2 to address your suggestions and the editors suggestions.